# Globally Soft Filter Pruning For Efficient Convolutional Neural Networks

## Abstract

This paper propose a cumulative saliency based Globally Soft Filter Pruning (GSFP) scheme to prune redundant filters of Convolutional Neural Networks (CNNs). Specifically, the GSFP adopts a robust pruning method, which measures the global redundancy of the filter in the whole model by using the soft pruning strategy. In addition, in the model recovery process after pruning, we use the cumulative saliency strategy to improve the accuracy of pruning. GSFP has two advantages over previous works: (1) **More accurate pruning guidance**. For a pre-trained CNN model, the saliency of the filter varies with different input data. Therefore, accumulating the saliency of the filter over the entire data set can provide more accurate guidance for pruning. On the other hand, pruning from a global perspective is more accurate than local pruning. (2) **More robust pruning strategy**. We propose a reasonable normalization formula to prevent certain layers of filters in the network from being completely clipped due to excessive pruning rate. Experiment results show that GSFP is effective on many classic CNN architectures and different data sets. Within my knowledge, GSFP is the first algorithm to combine global and soft pruning strategies. Notably, on MNIST and CIFAR-10, it achieves a much higher compression ratio compared with prior work while maintaining the same test accuracy.

## 1 Introduction

In recent years, convolutional neural networks (CNNs) have developed rapidly and achieved remarkable performance, which is widely applied in the field of computer vision, natural language processing, etc. However, CNNs with high computational and storage volumes, in fact, are difficult to deploy on embedded systems. As a result, model compression for CNNs have received hot attention.

Pruning-based approaches aim to remove the unnecessary connections of the neural network. parameter pruning which include non-structured pruning and structured pruning, reduces network complexity by directly reducing the scales of weight. Non-structured pruning (Han et al. (2015);Guo et al. (2016)) randomly deletes redundant neurons reducing model scales greatly. However, it is necessary to store additional coordinate information, resulting in a weak advantage in reducing compuational effort. Structured pruning (Wen et al. (2016);Mao et al. (2017);Luo et al. (2017);Zhu & Gupta (2018);He et al. (2018)) takes into account both the amount of computation and storage required. It performs regular model clipping based on filter, channel, and even a layer. Since the model pruned has good regularity, it can significantly reduce the scales of calculation and storage at the same time, which has recently received extensive attention. Essentially, the work of this paper is based on the idea of filter pruning.

In terms of Filter pruning, the current algorithm can be divided into "Soft" and "Hard" according to the training strategy, or "Local" and "Global" according to the pruning range. "Hard" is divided into two cases. One is that each iteration performs a pruning operation, so the pruning is too frequent to allow enough time recovering the model capacity. The second is to manually cut off the gradient update after pruning and then retraining(Luo et al. (2017), Dong et al. (2017), Li et al. (2017)). "Soft" means that each epoch performs a pruning operation, and the clipped filter can be updated during the retraining process(He et al. (2018)). "Local" means to set the pruning rate for each layer(He et al. (2017), Mao et al. (2017)), and "Global" means that only one overall pruning rate

is set, and the pruning rate of each layer is obtained by dynamic adjustment(Lin et al. (2018)). Essentially, the work of this paper is based on the pruning strategy of global and soft.

Most of the previous works on filter pruning, especially with the global pruning strategy, is poorly robust. That is, the distribution of pruning rate in each layer is extremely uneven. The filters of some layers are cut off a lot, while some layers are hardly cut. In extreme cases, when setting a too high pruning rate, the global pruning strategy will cause the filters of a certain layer removed completely, resulting in low accuracy. For example, analyzing the pruning results of VGG-16 in Li et al. (2017) did not trim the third to seventh convolutional layers, while the percentage of FLOPs in other convolutional layers was reduced by more than $50\%$ on CIFAR-10, as shown in Table 3. In Lin et al. (2018) the overall VGG-16 pruning rate is about $50\%$ on ImageNet, but the CONV3_3 layer pruned FLOPs is as high as $98\%$. In extreme cases, the pruning results of VGG-16 on CIFAR-10 are shown in Table 1, both the GHFP and the GSFP methods produce a low accuracy setting the pruning rate of $61\%$, which is the result of the removal of certain layers. Conversely, in Mao et al. (2017) , the imbalanced pruning results is prevented by setting the pruning rate of each layer. But this requires workers to have good prior knowledge and it takes a lot of time to perform repeated experiments.

In this work, we made efforts to improve the robustness of the global filter pruning algorithm. We proposed a cumulative saliency based Globally Soft Filter Pruning (GSFP) scheme, which globally prunes the unsalient filters and has better robustness. The contributions of this work are as follows:

- we experimentally compared the effects of four filter pruning strategies (global and local, hard and soft) on the results. And we found that combining global pruning with soft pruning strategy will have better results.

- we proposed a reasonable normalized saliency formula to improve the robustness of global pruning method.

- By accumulating saliency score across all batches, we obtain more accurate pruning guidance, as we restore the model capacity during retraining process.

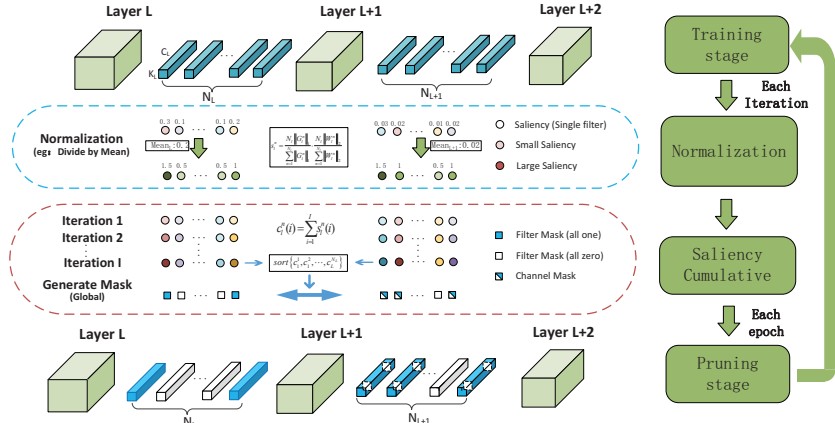

Figure 1: An illustration of GSFP. Each circle with a color represents the saliency score of a filter, and the rectangle represents the generated filter mask. The schematic diagram is divided into four parts: the top represents the training process and the bottom represents the pruning process. The middle blue dashed box mainly illustrates the use of normalized saliency formula to eliminate saliency scores uneven between different layers. The red dashed box shows the process of saliency accumulation and generation filter masks by global ordering.

The flowchar of proposed GSFP scheme is shown in Figure 1. In particular, we first got the saliency score of each filter through the saliency formula and normalized the saliency score of each filter by dividing the average saliency score of each layer. Then, accumulate the scores through the rebust saliency formula during each training iteration. After obtaining the cumulative saliency score, we sort the scores across all layers, and generate masks based on the prune rate to multiply the corresponding filters to complete the pruning operation. Finally, we iteratively tune the sparse network and dynamically update the filter saliency in a top-down manner.

## 2 MOTIVATION

In section 1, we explain that the current filter pruning algorithm can be divided into "Soft" and "Hard", or "Local" and "Global". In order to spy on the impact of different pruning strategies on the results, we compared the combined effects of the above four pruning strategies. As shown in Table 1, We test in extreme sparsity levels (Pruned FLOPS $61\%$ on VGG-16 in CIFAR-10). "L" refers to "Local", "G" refers to "Global", "H" refers to "Hard", "S" refers to "Soft", "F" refers to "Filter" and "P" refers to "Pruning". Therefore, "LHFP" means Locally Hard Filter Pruning method, and so on. The GHFP[1] method and the GHFP[2] method have the difference in the saliency score formula, and GHFP[2] uses a normalized score formula to achieve better robustness. This also applies to GSFP[1] and GSFP[2]. GSFP[3] uses cumulative normalized saliency formula to achieve more accurate pruning guidance compared to GSFP[2].

| Method | Pruned FLOPs | Acc | Local or Global | Hard or Soft | Score Formula |
|---|---|---|---|---|---|
| LHFP(Li et al. (2017)) | 61% | 92.15% | Local | Hard | $\|\boldsymbol{W}\|_2$ |
| LSFP(He et al. (2018)) | 61% | 92.44% | Local | Soft | $\|\boldsymbol{W}\|_2$ |
| GHFP[1](Lin et al. (2018)) | 61% | 10.00% | Global | Hard | $\|\boldsymbol{G}\cdot\boldsymbol{W}\|_1$ |
| GHFP[2](Molchanov et al. (2017)) | 61% | 92.76% | Global | Hard | Normalized |
| GSFP[1](ours) | 61% | 10.00% | Global | Soft | $\|\boldsymbol{G}\cdot\boldsymbol{W}\|_1$ |
| GSFP[2](ours) | 61% | 92.97% | Global | Soft | Normalized |
| GSFP[3](ours) | 61% | 93.29% | Global | Soft | Cumulative Normalized |

Table 1: Pruning results on VGG-16 in CIFAR-10 and comparisons to other approaches. Here, "Local" refers to setting the pruning rate for each layer, and "Global" refers to setting the global pruning rate from the entire model. "Hard" refers to pruning once per iteration or pruned without gradient update, "Soft" refers to pruning once per epoch and the pruned filters can be updated when training the model after pruning. "Normalized" formula reference formula 5(The GHFP[2] Normalized formula has slightly difference) and "Cumulative Normalized" formula reference formula 6.

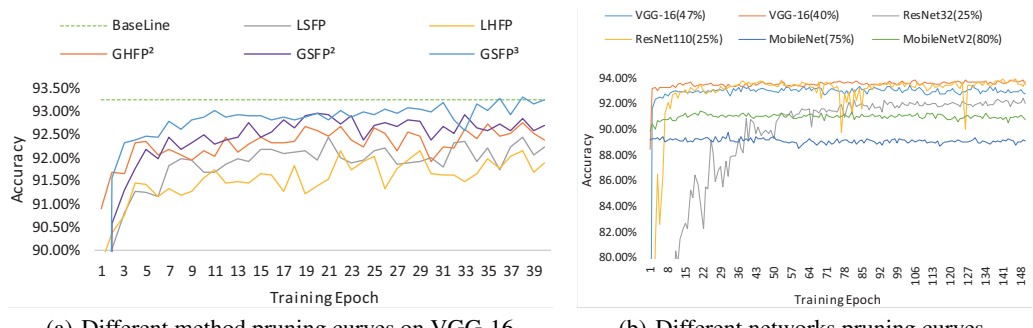

(a) Different method pruning curves on VGG-16.    (b) Different networks pruning curves.

Figure 2: Different method and networks pruning training curves on CIFAR-10.

The comparison results are shown in Table 1 and Figure 2(a). We have found:

- soft pruning strategy is better than hard, because the soft method has enough time to recover model capacity after pruning. This is clearly shown in Figure 2(a). Correspondingly, soft method also takes longer training time than hard.

- global pruning has worse robustness. As shown in Table 1, the accuracy drops rapidly under extreme sparsity on GHFP[1] and GSFP[1] compared with LHFP and LSFP. Because the filters of entire layer is comletely cut off. One of the reasons is that the size of the filter is not considered in the saliency formula. Though certain normalization methods, the robustness of global pruning can be improved.

- when the algorithm has better robustness, the global pruning can reach higher accuracy than the local. This can be obtained by comparing GHFP[2] and LHFP, GSFP[2] and LSFP.

In addition, we also found that cumulative saliency has more accurate pruning guidance. By accumulating saliency score across batches, GSFP[3] has achieved the best performance.

# 3 METHOD

## 3.1 PRELIMINARIES

We will formally introduce the symbol and annotations in this section. Consider a set of training examples $\mathcal{D} = \{\mathcal{X} = \{X_1, X_2, \cdots, X_N\}, \mathcal{Y} = \{Y_1, Y_2, \cdots, Y_N\}\}$, where $X_i$ and $Y_i$ represent an input and a target output, respectively. We denote filters of the entire network as $\mathcal{W} = \left\{W_1^1, W_1^2, \cdots, W_L^{N_L}\right\}$, where $\left\{W_l^n \in \mathbb{R}^{C_l \times K \times K}, 1 \le l \le L, 1 \le n \le N_l\right\}$. $C_l$ denotes the number of channels for the $l$-th convolution layer; $L$ denotes the number of layers; $N_l$ denotes the number of filters for the $l$-th convolution layer. During the training, the gradient value of the parameter is defined as $\mathcal{G} = \left\{G_1^1, G_1^2, \cdots, G_L^{N_L}\right\}$. The shapes of input tensor $U$ and output tensor $V$ are $N_l \times H_l \times W_l$ and $C_{l+1} \times H_{l+1} \times W_{l+1}$, respectively. The saliency score corresponding to each filter as $\mathcal{S} = \left\{s_1^1, s_1^2, \cdots, s_L^{N_L}\right\}$. By sorting the calculated saliency scores, we can get the mask of each filter according to the global pruning rate $\mathcal{P}$. Noted that mask $\mathcal{M} = \left\{M_1^1, M_1^2, \cdots, M_L^{N_L}\right\}$ has two forms corresponding to filter mask and channel mask in Figure 1. The parameters after pruning are defined as $\mathcal{W}^* = \left\{W_1^{1*}, W_1^{2*}, \cdots, W_L^{N_L*}\right\}$.

In the pruning process, we hope to replace the original network parameters with more sparse and regular parameters to achieve a trade-off between reducing the amount of calculation and not losing accuracy as $\mathcal{L}(\mathcal{D}|\mathcal{W}^*) = \mathcal{L}(\mathcal{D}|\mathcal{W} \cdot \mathcal{M}) \approx \mathcal{L}(\mathcal{D}|\mathcal{W})$, where $\mathcal{L}(\cdot)$ is a loss function for the pruned network. The pruning problem translates into the following optimization problem:

$$
\begin{aligned}
min \quad & \mathcal{L}(\mathcal{D}|\mathcal{W}^*) \\
s.t. \quad & \mathcal{W}^* = \mathcal{W} \cdot \mathcal{M} \\
& \|\mathcal{M}\|_0 \le \mathcal{P} \sum_{l=1}^{L} N_l
\end{aligned}
\tag{1}
$$

where $\|\mathcal{M}\|_0$ is the number of filter pruning form(all $M$ internal elements are zero) in $\mathcal{M}$. In response to these problems, the key is to find a suitable mask. Therefore, we need to develop a set of rules to evaluate the importance of each filter.

## 3.2 GLOBALLY SOFT FILTER PRUNING (GSFP)

Most of previous works(Li et al. (2017);Luo et al. (2017); Lin et al. (2018)) pruned the filter every training iteration. This pruning strategy leads to a high rate of mis-pruning, and the model after pruning recovery time is short so that it will reduce the capacity. In addition, the constructed saliency formula either does not take into account global redundancy or ignores normalization of each layer. This work proposed a more reasonable saliency formula to make the pruning strategy more robust and improved the accuracy of pruning by accumulating saliency scores. The details of GSFP is illustratively explained in Figure 1, which can be divided into the following three steps.

**Rebust Saliency Formula.** Some works such as Li et al. (2017) defined the saliency formula as the sum of the absolute values of kernel weights. He et al. (2018) used the $l_2$-norm to evaluate the importance of each filter. Molchanov et al. (2017) and Lin et al. (2018) proposed that not only the weight but also the gradient value affecting the loss function, which is derived by Taylor expansion.

$$
|\Delta \mathcal{L}(\mathcal{W}^*)| = \mathcal{L}(\mathcal{D}|\mathcal{W}^*) - \mathcal{L}(\mathcal{D}|\mathcal{W}) \approx \left| \frac{\partial \mathcal{L}(\mathcal{D}|\mathcal{W}^*)}{\partial \mathcal{W}^*} \cdot \mathcal{W}^* \right|
\tag{2}
$$

However when the global pruning strategy is adopted, the pruning result will be uneven if the weights between the different layers are very different. Obviously, the extremely uneven pruning result is an important cause of poor pruning robustness. We all know that filter sizes are often different in different convolution layers. However, the size of the filter is not considered in the formula which is one of the reasons for the uneven pruning results. So we have modified the $l_2$-norm formula as follows:

$$
\|W_l^n\|_2 = \sqrt{\frac{1}{C_l \cdot K \cdot K} \sum_{c=1}^{C_i} \sum_{k_1=1}^{K} \sum_{k_2}^{K} |w_l^n(c, k_1, k_2)|^2}
\tag{3}
$$

In order to highlight the saliency of the filter for different input data, we use the gradient value in back propagation as the multiplier factor. The specific definition is as follows:

$$\|\boldsymbol{G}_l^n\|_1 = \frac{1}{C_l \cdot K \cdot K} \cdot \sum_{c=1}^{C_i} \sum_{k_1=1}^{K} \sum_{k_2}^{K} |g_l^n(c, k_1, k_2)| \tag{4}$$

Different convolutional layers have different parameter distributions, and they also accompany vanishing or exploding gradient problem during training. Taking into account these two phenomena, we normalize the gradient factor and the weighting factor separately, and finally construct a complete saliency formula as follows:

$$s_l^n = \frac{N_l \|\boldsymbol{G}_l^n\|_1}{\sum_{n=1}^{N_l} \|\boldsymbol{G}_l^n\|_1} \cdot \frac{N_l \|\boldsymbol{W}_l^n\|_2}{\sum_{n=1}^{N_l} \|\boldsymbol{W}_l^n\|_2} \tag{5}$$

**Cumulative Saliency.** In order to improve the accuracy of pruning, we accumulate the saliency scores of the filters in each batch during the training process. During the training stage, the saliency score is calculated while restoring the capacity of the model. The saliency score continues to accumulate in each iteration until the entire data set is traversed and then prune based on the saliency score, as shown in Figure 1. We define the cumulative significance score as $\mathcal{C} = \left\{ c_1^1, c_1^2, \cdots, c_L^{N_L} \right\}$. The process of saliency accumulation is described as:

$$c_l^n = \sum_i^I s_l^n(i) \tag{6}$$

where $I$ represents the number of iterations within a training epoch. This takes into account the saliency of the filter throughout the data set and provides more accurate guidance for subsequent pruning processing.

**Filter Pruning.** In the filter pruning step, we use a global pruning strategy to fully consider the redundancy of each layer. We only need to set a global pruning rate $\mathcal{P}$. The global saliency scores of all filters $Indx$ is constructed, which are sorted by a descending order, i.e., $Indx = sort(\left\{ c_1^1, c_1^2, \cdots, c_L^{N_L} \right\})$. Then set the top-$\mathcal{P} \sum_{l=1}^{L} N_l \, Indx$ corresponding filters to zero. Note that if we set the value of selected $N_l \mathcal{P}_l$ filters to zero, the channel corresponding to the next layer of filters will be set zero simultaneously, i.e., $\left\{ \boldsymbol{W}_{l+1}^n \in \mathbb{R}^{N_l \times (1-\mathcal{P}_l) \times K \times K}, 1 \le n \le N_{l+1} \right\}$.

## 4 EXPERIMENTS

To evaluate the effectiveness of the GSFP scheme, CNN model pruning were performed on three databases MNIST, CIFAR-10, and ImageNet. We implemented our filter pruning method in the PyTorch framework. To get the baseline accuracies for each network, we train each model from scratch on MNIST and CIFAR-10. For the ImageNet dataset, pre-trained model from torchvision were used as the baseline model. The same data argumentation strategies were used with PyTorch official examples (Paszke et al. (2017)). For retraining of filter pruning, we use a constant learning rate 0.01 and retrained 100 epochs on MNIST. In CIFAR-10, we set the initial learning rate to 0.01, multiply by 0.1 per 50 epoch, and retrained 150 epochs. Finally, we retrained 100 epochs on ImageNet. The first 70 epochs learning rates are set to 0.001, and the last 30 epochs settings are 0.0001.

### 4.1 LeNet on MNIST

MNIST is a well-known database of handwritten digits which contains about $60,000$ images for training and $10,000$ for testing. We perform experiments on a version of LeNet proposed in (LeCun et al. (1998)), which consists of two convolutional and two fully-connected layers. When applying GSFP to LeNet, we first need a pre-trained model and then pruning filter based on saliency score for each training epoch. Table 2 summarizes the remained filters and channels, floating-point operations (FLOPs), and classification accuracy. Compared with the SSL method (Wen et al. (2016)), GSFP can prune more filters and channels resulting in a number of $3\%$ reduction on computation complexity on average, while maintaining the same accuracy.

| Method | Filters | Channel | FLOPs | Error |
|---|---|---|---|---|
| Baseline | 20 — 50 | 1 — 20 | 100% — 100% | 0.75% |
| Wen et al. (2016) | 5 — 19 | 1 — 4 | 25% — 7.6% | **0.80%** |
| Wen et al. (2016) | 3 — 12 | 1 — 3 | 15% — 3.6% | **1.00%** |
| GSFP (70%) | 7 — 14 | 1 — 7 | 35% — 9.8% | 0.72% |
| GSFP (80%) | 6 — 8 | 1 — 6 | 30% — 4.8% | 0.77% |
| GSFP (85%) | 5 — 6 | 1 — 5 | 25% — 3.0% | **0.83%** |
| GSFP (90%) | 3 — 4 | 1 — 3 | 15% — 1.2% | **1.07%** |

Table 2: Results after pruning unimportant filters and channels in LeNet-5

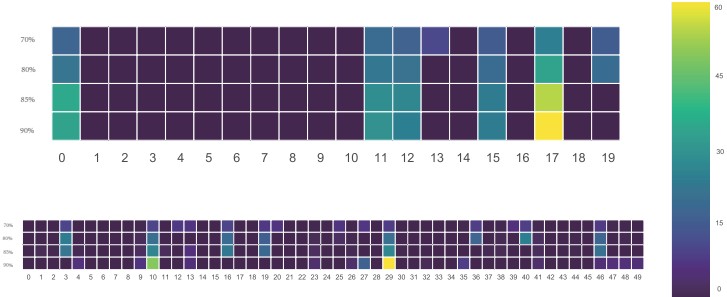

Figure 3: The saliency scores visualization for LeNet-5 on 10 epoch. The top figure shows saliency scores for CONV1, bottom figure shows the saliency scores for CONV2. The pruning rate are 70%, 80%, 85% and 90%, respectively.

To further analyze the significance of the saliency score for pruning, we visualized the saliency scores for the four pruned models with different pruning rates in Figure 3. In the figure, the 17th and 29th filters are critical filter. We can see that as the pruning rate increases, the saliency of the critical filter becomes more and more prominent during the process of model retraining. When we set the pruning rate to 70%, 80%, 85% and 90%, respectively, the corresponding convolution kernel of CONV1 layer remained is 7, 6 , 5 , 3 in Table 2. In Figure 3, we can clearly gauge the importance of each filter across the network. When the pruning rate is set to 90%, CONV1 can only retain 3 filters after global measurement, but as can be seen from the figure 3 , there are 5 filters that are obviously saliency, we need drop two important filters, so the classification accuracy drop about 0.32%. Therefore, visual saliency scores can help us find the most appropriate pruning rate.

| Layer Type | VGG-16 Orig | | Li et al. (2017) | | | GSFP | | |
|---|---|---|---|---|---|---|---|---|
| | Params | MFLOPs | Params | MFLOPs | Pruned | Params | MFLOPs | Pruned |
| Conv2d-1 | 1.73E+03 | 1.77 | 8.64E+02 | 0.88 | 50% | 5.94E+02 | 0.61 | 65.62% |
| Conv2d-2 | 3.69E+04 | 37.75 | 1.84E+04 | 18.87 | 50% | 8.51E+03 | 8.72 | 76.90% |
| Conv2d-3 | 7.37E+04 | 18.87 | 7.37E+04 | 18.87 | 0% | 3.25E+04 | 8.32 | 55.91% |
| Conv2d-4 | 1.47E+05 | 37.75 | 1.47E+05 | 37.75 | 0% | 7.79E+04 | 19.93 | 47.19% |
| Conv2d-5 | 2.95E+05 | 18.87 | 2.95E+05 | 18.87 | 0% | 1.94E+05 | 12.40 | 34.30% |
| Conv2d-6 | 5.90E+05 | 37.75 | 5.90E+05 | 37.75 | 0% | 3.50E+05 | 22.39 | 40.68% |
| Conv2d-7 | 5.90E+05 | 37.75 | 5.90E+05 | 37.75 | 0% | 3.21E+05 | 20.57 | 45.51% |
| Conv2d-8 | 1.18E+06 | 18.87 | 5.90E+05 | 9.44 | 50% | 4.89E+05 | 7.82 | 58.54% |
| Conv2d-9 | 2.36E+06 | 37.75 | 5.90E+05 | 9.44 | 75% | 4.84E+05 | 7.74 | 79.49% |
| Conv2d-10 | 2.36E+06 | 37.75 | 5.90E+05 | 9.44 | 75% | 4.10E+05 | 6.57 | 82.60% |
| Conv2d-11 | 2.36E+06 | 9.44 | 5.90E+05 | 2.36 | 75% | 4.84E+05 | 1.94 | 79.49% |
| Conv2d-12 | 2.36E+06 | 9.44 | 5.90E+05 | 2.36 | 75% | 4.48E+05 | 1.79 | 81.03% |
| Conv2d-13 | 2.36E+06 | 9.44 | 5.90E+05 | 2.36 | 75% | 4.82E+05 | 1.93 | 79.59% |
| Linear-14 | 5.12E+03 | 0.01 | 5.12E+03 | 0.01 | 0% | 4.79E+03 | 0.01 | 6.45% |
| Total | 1.47E+07 | 313.20 | 5.26E+06 | 206.15 | **34%** | 3.79E+06 | 120.732 | **61.45%** |

Table 3: VGG-16 pruning model detail comparison on CIFAR-10.

## 4.2 RESULT ON CIFAR-10

For CIFAR-10 dataset, we have tested our GSFP on single-branch networks(VGG-16), multi-branch networks(ResNet-32 and ResNet-110) and compact networks(MobileNet and MobileNet-V2). We use several difference pruning rates to select the best trade-off between model computation and classification accuracy. Table 4 summaries the pruning results and Figure 2(b) shows the training curve for different networks.

For example, in the case where the same classification without loss of accuracy for single-branch networks VGG-16 (Simonyan & Zisserman (2014)), (Li et al. (2017)) pruning method reduced FLOPs by 34.2%, while GSFP reduced FLOPs by 61.46%. The proposed scheme have achieved 2× better results. Note that VGG-16 on CIFAR-10 consists of 13 convolutional layers adding Batch Normalization and just 1 fully connected layer, in which the fully connected layers do not occupy large portions of parameters due to the small input size and less hidden units. For single-branch networks we adopted a strategy of combining filter pruning with channel pruning. Because the previous layer of filter pruning will affect the next layer of convolution filter channels. Detailed pruning results are shown in Table 3. Compared with the pruning results listed in Li et al. (2017), the pruning results of the GSFP are more balanced in each layer, which is why the GSFP can pruning more, similar to the buckets effect.

For multi-branch networks such as ResNet-32 and ResNet-110 models (He et al. (2016)) on CIFAR-10, we need to consider the shortcut connection structure in the process of pruning. Since there is a cross-layer connection for each residual block of ResNet, we only perform filter pruning for the first layer convolution and the shortcut convolution layer of each block, and other layer filter pruning combined with channel pruning. The proposed method reduced FLOPs by 50% on ResNet-32 without losing the accuracy of the pre-trained model. With the same recognition accuracy, the proposed sheme achieved 7% more pruning rates compared to He et al. (2018). For the ResNet-110, our scheme has a advantage of 12% on pruning rates compared with the best result of the state-of-the-art.

Moreover, we have also pruned two compact networks MobileNet (Howard et al. (2017)) and MobileNetV2 (Sandler et al. (2018)). Compact network usually uses group convolution unit to reduce the amount of calculations and parameters, so we only conducted filter pruning for these networks. After pruning, it is found that MobileNet and MobileNetV2 have a lot of redundancy on the CIFAR-10 dataset, the amount of calculation pruning can reach 72.57% and 69.68%, respectively.

| Model | Method | MFLOPs | Pruned FLOPs | Acc |
|---|---|---|---|---|
| | BaseLine | 313.2 | – | 93.25% |
| | Li et al. (2017) | 206.1 | 34.20% | 93.40% |
| VGG-16 | GSFP (40%) | 167.3 | 46.58% | 93.83% |
| | GSFP (45%) | 134.2 | 57.15% | **93.43%** |
| | GSFP (47%) | 120.7 | **61.46%** | 93.29% |
| | BaseLine | 68.9 | – | 92.63% |
| | Dong et al. (2017) | 47.0 | 31.79% | 90.74% |
| ResNet-32 | He et al. (2018) | 40.3 | 41.51% | 92.08% |
| | GSFP (25%) | 35.8 | 48.05% | **92.63%** |
| | GSFP (27%) | 34.4 | **50.02%** | 92.13% |
| | BaseLine | 252.9 | – | 93.68% |
| | Dong et al. (2017) | 166.4 | 34.20% | 93.44% |
| ResNet-110 | Li et al. (2017) | 155.3 | 38.59% | 93.30% |
| | He et al. (2018) | 149.7 | 40.81% | 93.86% |
| | GSFP (25%) | 120.1 | **52.51%** | **93.90%** |
| | BaseLine | 46.4 | – | 89.63% |
| MobileNet | GSFP (75%) | 14.1 | 69.64% | **89.74%** |
| | GSFP (78%) | 12.7 | **72.57%** | 89.41% |
| | BaseLine | 91.2 | – | 91.51% |
| MobileNet-V2 | GSFP (80%) | 40.0 | 56.17% | **91.44%** |
| | GSFP (85%) | 27.6 | **69.68%** | 90.89% |

Table 4: Comparison of pruning result on CIFAR-10. The MFLOPs denotes million floating-point operations. The values in parentheses represent the global filter pruning ratio.

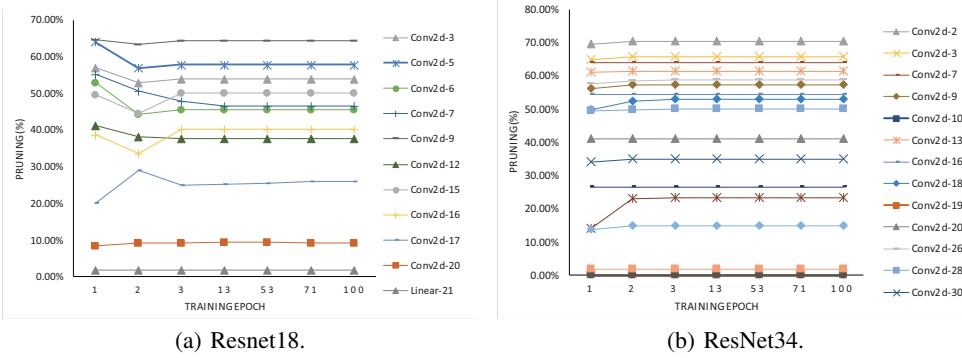

(a) Resnet18.  (b) ResNet34.

Figure 4: Different layers of pruning distribution curves on ImageNet

## 4.3 RESNET ON IMAGENET

We also test our pruning scheme on the large scale ImageNet classification task. ResNet is currently the most widely used network structure, so in the experiment we chose ResNet-18 and ResNet-32 for pruning. During the pruning process, since the ResNet cross-layer contains a lot of information, we do not pruned the shortcut convolution layer. Table 5 shows the pruning results of ResNet on Imagenet. For ResNet-18, GSFP reduced the FLOPs by $47.06\%$, with a Top-1 accuracy drop $2.95\%$. For ResNet-34, under almost the same loss of precision, the GSFP was reduced by 3.75% compared to the He et al. (2018). In addition, when the pruning was further increased, the calculation amount was reduced by 9.6%, while the Top1 precision loss was only decreased by 0.5%. Compared to Dong et al. (2017) , Li et al. (2017) and He et al. (2018), our proposed GSFP has a larger pruning space. By analyzing the pruning results of convolution layers at different time points in Figure 4, we can see that the strategy of globally soft pruning can automatically adjust the pruning result and improve the robustness of the pruning process.

| Model | Method | GFLOPs | Pruned | Acc BaseLine | Acc After Pruned | Acc Drop |
|---|---|---|---|---|---|---|
| | Dong et al. (2017) | 1.18 | 34.60% | 69.98% / 89.24% | 66.33% / 86.94% | 3.06% / 2.30% |
| ResNet-18 | He et al. (2018) | 1.05 | 41.80% | 70.28% / 89.63% | 67.10% / 87.78% | 3.18% / 1.85% |
| | GSFP (20%) | 0.96 | **47.06%** | 69.76% / 89.08% | 66.81% / 87.07% | **2.95%** / 2.01% |
| | Dong et al. (2017) | 2.75 | 24.80% | 73.42% / 91.36% | 72.99% / 91.19% | 0.43% / 0.17% |
| | Li et al. (2017) | 2.76 | 24.20% | 73.23% / – | 72.17% / – | 1.06% / – |
| ResNet-34 | He et al. (2018) | 2.16 | 41.10% | 73.92% / 91.62% | 71.83% / 90.33% | 2.09% / 1.29% |
| | GSFP (20%) | 2.02 | 44.85% | 73.30% / 91.42% | 71.16% / 90.06% | 2.14% / 1.36% |
| | GSFP (25%) | 1.80 | **50.74%** | 73.30% / 91.42% | 70.70% / 89.67% | 2.60% / 1.75% |

Table 5: Comparison of pruning results on ImageNet. The GFLOPs is the giga floating-point operations. The Acc Drop is the accuracy of the pruned model minus that of the baseline model, so a smaller number of Acc Drop is better.

## 5 CONCLUSIONS

In this work, we propose a scheme GSFP on filter pruning for convolutional neural networks, which combines global pruning strategy and soft pruning strategy. First, we designed the experiment and compared the previous work with four different pruning strategies (global and local, hard and soft). In addition, it is proved by experiments that the global pruning strategy will reduce the pruning robustness and can be improved by a reasonable normalization formula. Finally, the pruning accuracy can be effectively improved by accumulating the saliency of the filter across all batches in the model recovery training process after pruning.

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
