# OpenReview forum: "Globally Soft Filter Pruning For Efficient Convolutional Neural Networks"
_ICLR.cc/2019/Conference_

### Official Review · AnonReviewer3 · 2018-10-29
**Weak reject: Incremental work and insufficient experiments.**

**Rating:** 4
**Confidence:** 3

**Review:**

This paper proposes new heuristics to prune and compactify neural networks. The heuristics try to consider 1) filter weight and gradient normalization by their size, 2) saliency normalization across layers, 3) saliency accumulation across batch. The author claims that these can address problems previous studies had and experimental results show that the proposed method achieve higher compression ration with less loss of accuracy.

This paper discusses how to determine the importance of filters. As cited in the paper, there have been various attempts to tackle the same problem and the paper contributes to the series of efforts. The paper introduces a new way to compute such importance values based on their observations. The method is tested on a few dataset and a various models and compared with some previous studies. I like the simple but yet effective method, however, I think it is not good enough for ICLR.

1. What is effective is not very clear.

The paper pointed out issues of previous studies and proposed the new method based on the observations. However, only the final method is compared with other work and it did not examine which part of the method was essential. The paper needs more detailed analyses on the proposed method. For example, the readers would want to know if the normalization in Eq. (2) is really important or not. The readers would be also interested in a visualization like Fig. 2 without saliency normalization.

2. The numbers of previous studies come only from their papers.

It is very difficult to know if the proposed method is actually better than the previous methods if the numbers just come from their papers. We want to compare the ideas, but not numbers. The essential ideas of other papers need to be abstracted and tested in the paper by itself. It relates to the first item above. "Baseline" should be a baseline method but not models without pruning.

Numbers from other papers are still useful to show that the numbers in the paper are good in an absolute manner.

3. Weak theoretical reasoning

Eq. (1) in the paper is not actually used for optimization while some previous methods do. If the proposed method is better than other methods which directly optimizes the loss, should we think that the formulation itself is bad?

The paper discusses imbalanced pruned pruning results. It needs to show that it is actually bad.

* minor things

** Table 1: Should the first row of "Wen et al. (2016)" have "5-19" and "1-5" or "4-19" and "1-4" for "Filters" and "Channels", respectively?

** I'd recommend another proofreading.

---

> ### Author Response · Authors · 2018-11-25
> **Reply to Reviewer3**
>
> Thank you for your detailed review, we have updated the paper to accommodate your suggested improvements.
>
> 1. Modification
>
>    1.1 We reorganized the content of this paper and added Motivation(Section 2). In section 2, we added a comparative experiment of pruning strategies for the VGG-16 on the CIFAR-10 dataset.  The design of the comparative experiment is mainly to abstract the ideas of the previous paper (including Hard pruning, Soft pruning, Local pruning and Global pruning) and compare them in combination. We added a description of the different pruning strategies in section 1 (Introduction). At the same time, we compare the proposed saliency formulas one by one and conclude that the cumulative normalized saliency formula is effective. Comparison results are shown in Table 1 and Figure 2(a).
>
>    1.2 Previously submitted paper did not provide a good illustration of the innovation of the algorithm. We rewrite section 1 and list the contributions of this paper one by one.
>
>    1.3 We modified section 3.2.1 to add a Taylor expansion derivation and reasonably explained the effect of gradient factors and weights factors on the loss function. This derived from "Pruning Convolutional Neural Networks for Resource Efficient Inference", Molchanov, et al., 2017 and "Accelerating convolutional networks via global & dynamic filter pruning."  Shaohui Lin, et al 2018
>
> 2. Experimental Result
>
> Through experiments, we can prove that global pruning is better than local pruning, soft pruning is better than hard pruning. Within my knowledge, GSFP is the first to combine global pruning strategy with soft pruning strategy. In addition to this, experiments prove that a reasonable and normalized saliency formula can improve the problem of extremely unbalanced pruning results(all the kernels of some layers are cut off) at high pruning rates. Finally, the accuracy of the pruning can be improved by accumulating the saliency score in the model recovery training process after pruning.
>
> 3. Other
>
> ** The GSFP algorithm proposed in this paper only normalized within the layer, because the saliency score is accumulated, so the results in Fig. 2 (Modified as Fig 3) will be significantly different.
>
> ** I have proofread the results of pruning in Wen et al (2016), which is indeed "5-19" and "1-4". Perhaps the filter pruning is separate from the channel pruning.

---

### Official Review · AnonReviewer1 · 2018-11-02
**Method for pruning networks with global network saliency computed for the entire dataset**

**Rating:** 5
**Confidence:** 4

**Review:**

This paper proposes a method for pruning CNNs, which considers all filters globally. It normalizes the weights of a filter within a layer and globally across all layers. To incorporate the effect of the data set, the authors additionally compute the normalized gradients and multiply the weight term with it. The third concept that the authors introduce is the idea of accumulating this measure of the importance of a filter for one entire epoch, before pruning.

The paper is missing an important citation: "Pruning Convolutional Neural Networks for Resource Efficient Inference", Molchanov, et al., 2017, which previously introduced many of the concepts proposed in this submission. For example, Molchanov, et al., propose the magnitude of the product of the activation and its gradient as the criteria for pruning. They derive mathematically as to how their criterion relates to the expected change in the overall loss via the Taylor series expansion. They additionally emphasize the importance of normalizing the criterion across network layers and the importance of greater number of updates to the network before pruning for better results. It is important that authors of this submission clearly compare and differentiate their work from the previous very closely related work of Molchanov et al.

Additionally, in order to understand the contribution of each of the concepts that the authors introduce, i.e., accumulation of saliency, multiplication with gradients, normalization within a layer, normalization across layers, the authors should present ablation studies to show the affect of each of these concepts independently to the overall accuracy of their approach.

---

> ### Author Response · Authors · 2018-11-25
> **Reply to Reviewer1**
>
> Thank you for your detailed review, we have updated the paper to accommodate your suggested improvements.
>
> 1. Modification
>
>    1.1 We reorganized the content of this paper and added Motivation(section 2). In section 2, we added a comparative experiment of pruning strategies for the VGG-16 on the CIFAR-10 dataset.  The design of the comparative experiment is mainly to abstract the ideas of the previous paper (including Hard pruning, Soft pruning, Local pruning and Global pruning) and compare them in combination. We added a description of the different pruning strategies in section 1 (Introduction). At the same time, we compare the proposed saliency formulas one by one and conclude that the cumulative normalized saliency formula is effective. Comparison results are shown in Table 1 and Figure 2(a).
>
>    1.2 Previously submitted paper did not provide a good illustration of the innovation of the algorithm. We rewrite section 1 and list the contributions of this paper one by one.
>
>    1.3 We modified section 3.2.1 to add a Taylor expansion derivation and reasonably explained the effect of gradient factors and weights factors on the loss function. This derived from "Pruning Convolutional Neural Networks for Resource Efficient Inference", Molchanov, et al., 2017 and "Accelerating convolutional networks via global & dynamic filter pruning."  Shaohui Lin, et al 2018
>
> 2. Experimental Result
>
> Through experiments, we can prove that global pruning is better than local pruning, soft pruning is better than hard pruning. Within my knowledge, GSFP is the first to combine global pruning strategy with soft pruning strategy. In addition to this, experiments prove that a reasonable and normalized saliency formula can improve the problem of extremely unbalanced pruning results(all the kernels of some layers are cut off) at high pruning rates. Finally, the accuracy of the pruning can be improved by accumulating the saliency score in the model recovery training process after pruning.
>
> 3. Other
>
> ** We added "Pruning Convolutional Neural Networks for Resource Efficient Inference", Molchanov, et al. to the experiment and compared it with the method in this paper.
>
> ** The GSFP algorithm proposed in this paper is only normalized within the layer,  and there is no normalization across layers.

---

> > ### Comment · AnonReviewer1 · 2018-11-27
> > **Response to authors**
> >
> > I thank the authors for addressing the various concerns raised by the reviewers and for performing additional experiments. I am satisfied with the authors experiments to demonstrate the superior performance of their proposed method versus the work of Molchanov et al., 2017. However, given how similar the two works are conceptually, which the authors have attempted to clarify in the revised paper, the novelty of the current work is marginal and mostly incremental. For that reason, I will maintain my previous rating.

---

### Official Review · AnonReviewer2 · 2018-11-06
**using cumulative saliency as guidance for model pruning**

**Rating:** 6
**Confidence:** 4

**Review:**

In this paper, the authors propose to use cumulative saliency as guidance for model pruning. In particular, when designing saliency, they introduce a balanced formula by taking the filter size and gradient value into account.  The paper is well organized, and extensive experiments are investigated.  However, the novelty is relatively limited. The advantage of the proposed method is marginal on ImageNet, when comparing with the relevant approaches.

---

> ### Author Response · Authors · 2018-11-25
> **Reply to Reviewer2**
>
> Thank you for your detailed review, we have updated the paper to accommodate your suggested improvements.
>
> 1. Modification
>
>    1.1 We reorganized the content of this paper and added Motivation(section 2). In section 2, we added a comparative experiment of pruning strategies for the VGG-16 on the CIFAR-10 dataset.  The design of the comparative experiment is mainly to abstract the ideas of the previous paper (including Hard pruning, Soft pruning, Local pruning and Global pruning) and compare them in combination. We added a description of the different pruning strategies in section 1 (Introduction). At the same time, we compare the proposed saliency formulas one by one and conclude that the cumulative normalized saliency formula is effective. Comparison results are shown in Table 1 and Figure 2(a).
>
>    1.2 Previously submitted paper did not provide a good illustration of the innovation of the algorithm. We rewrite section 1 and list the contributions of this paper one by one.
>
>    1.3 We modified section 3.2.1 to add a Taylor expansion derivation and reasonably explained the effect of gradient factors and weights factors on the loss function. This derived from "Pruning Convolutional Neural Networks for Resource Efficient Inference", Molchanov, et al., 2017 and "Accelerating convolutional networks via global & dynamic filter pruning."  Shaohui Lin, et al 2018
>
> 2. Experimental Result
>
> Through experiments, we can prove that global pruning is better than local pruning, soft pruning is better than hard pruning. Within my knowledge, GSFP is the first to combine global pruning strategy with soft pruning strategy. In addition to this, experiments prove that a reasonable and normalized saliency formula can improve the problem of extremely unbalanced pruning results(all the kernels of some layers are cut off) at high pruning rates. Finally, the accuracy of the pruning can be improved by accumulating the saliency score in the model recovery training process after pruning.
>
> 3. Other
>
> ** For ResNet-18 , GSFP reduced the FLOPs by 47.06% , with a Top-1 accuracy drop 2.95%. For ResNet-34, under almost the same loss of precision, the GSFP was reduced by 3.75% compared to the "Soft filter pruning for accelerating deep convolutional neural networks" He  et al. In addition, when the pruning was further increased, the calculation amount was reduced by 9.6%, while the Top1 precision loss was only decreased by 0.5%.  I think the method proposed in this paper is effective, but it is more obvious on the CIFAR-10 and MNIST data sets.

---

### Meta-Review · Area_Chair1 · 2018-12-17
**not convincing**

**Confidence:** 4
**Recommendation:** Reject

**Metareview:**

This paper proposes new heuristics to prune and compress neural networks. The paper is well organized. However, reviewers are concerned that the novelty is relatively limited. The advantage of the proposed method is marginal on ImageNet. What is effective is not very clear. Therefore, recommend for rejection.